# Fire air pollution reduces global terrestrial productivity

Xu Yue [1] & Nadine Unger [2]

Fire emissions generate air pollutants ozone ($O_3$) and aerosols that influence the land carbon cycle. Surface $O_3$ damages vegetation photosynthesis through stomatal uptake, while aerosols influence photosynthesis by increasing diffuse radiation. Here we combine several state-of-the-art models and multiple measurement datasets to assess the net impacts of fire-induced $O_3$ damage and the aerosol diffuse fertilization effect on gross primary productivity (GPP) for the 2002–2011 period. With all emissions except fires, $O_3$ decreases global GPP by $4.0 \pm 1.9$ Pg C yr$^{-1}$ while aerosols increase GPP by $1.0 \pm 0.2$ Pg C yr$^{-1}$ with contrasting spatial impacts. Inclusion of fire pollution causes a further GPP reduction of $0.86 \pm 0.74$ Pg C yr$^{-1}$ during 2002–2011, resulting from a reduction of $0.91 \pm 0.44$ Pg C yr$^{-1}$ by $O_3$ and an increase of $0.05 \pm 0.30$ Pg C yr$^{-1}$ by aerosols. The net negative impact of fire pollution poses an increasing threat to ecosystem productivity in a warming future world.

[1] Climate Change Research Center, Institute of Atmospheric Physics, Chinese Academy of Sciences, Beijing 100029, China. [2] College of Engineering, Mathematics and Physical Sciences, University of Exeter, Exeter EX4 4QE, UK. Correspondence and requests for materials should be addressed to X.Y. (email: yuexu@mail.iap.ac.cn) or to N.U. (email: N.Unger@exeter.ac.uk)

Fire is an important disturbance to the terrestrial carbon budget. Every year, global fires directly emit 2–3 Pg (=$10^{15}$ g) carbon into the atmosphere[1]. This immediate carbon loss is partially compensated by an enhancement in net ecosystem productivity driven by changes in canopy composition and soil respiration[2]. In addition to the carbon emissions, fire plumes also generate short-lived climate pollutants, including ozone ($O_3$) and fine mode aerosols (e.g., $PM_{2.5}$, particulate matter less than 2.5 μm in diameter), which may worsen air quality in the local and downwind regions[3]. Globally, fires contribute to 3.5% of the total tropospheric $O_3$[4] and 16% of total aerosol optical depth (AOD)[5]. Regionally, fires can enhance surface $O_3$ by 10–30 ppbv[6,7] during pollution episodes and dominate $PM_{2.5}$ with average contributions of 30–55%[8,9] during fire seasons.

While the impacts of fire $O_3$ and aerosols to public health have been widely recognized and studied[10,11], the ecological impacts of this air pollution on ecosystem health and the land carbon cycle lack systematic quantification. Increases in $O_3$ and aerosols have strongly contrasting impacts on plant productivity[12]. $O_3$ is phytotoxic and reduces plant photosynthesis[13–15]. Aerosol pollution may promote photosynthesis by enhancing diffuse radiation[16,17], and exert varied impacts on land carbon uptake through concomitant perturbations in meteorology[18–20]. Limited studies have accounted for the fire air pollution impacts on regional land carbon assimilation[21,22], and shown that fire $O_3$ and aerosols result in comparable but opposite perturbations in vegetation productivity over the Amazon Forest. However, the combined effects of $O_3$ and aerosols from the same fires remain unquantified at both the regional and global scales.

Here, we examine the net impacts of fire $O_3$ and aerosols on ecosystem productivity using a suite of validated models in combination with multiple measurements (Methods). The chemical transport model GEOS-Chem[23] is used to predict fire-induced changes in aerosol optical depth and surface $O_3$. The Column Radiation Model (CRM) is applied to quantify perturbations in diffuse and direct solar radiation caused by fire aerosols. The Yale Interactive terrestrial Biosphere (YIBs) model[24], a dynamic global vegetation model, is used to quantify changes in ecosystem GPP due to $O_3$ inhibition and aerosol diffuse fertilization effects (DFE) originating from fire pollution. For this study, we do not consider ecosystem responses to aerosol-induced perturbations in meteorology and cloud due to the large uncertainties[19]. Joint simulations reveal an indirect terrestrial carbon loss caused by fire air pollutants, because fire $O_3$ strongly dampens ecosystem productivity of unburned forests and masks the benefit of increased diffuse radiation from fire aerosols.

## Results

### Evaluation of GPP responses to $O_3$ and diffuse radiation.
The YIBs model calculates $O_3$ damages to GPP as a function of stomatal $O_3$ flux[25], which is dependent on both $O_3$ concentrations ([$O_3$]) and leaf stomatal conductance (Methods). Compared with an observation-based meta-analysis (Supplementary Data 1), the model reasonably captures GPP sensitivity to enhanced [$O_3$] for most plant functional types (PFTs), especially for the temperate deciduous trees, C4 herbs, and crops (slopes of lines, Fig. 1). For these PFTs, GPP is found to decrease by 0.24–0.30% for a 1 ppbv [$O_3$] increase. The model underestimates the GPP sensitivity by 62.5% for temperate evergreen trees while overestimates the GPP response by 37.5% for evergreen broadleaf trees compared with available measurements. Observed GPP–$O_3$ relationships show large uncertainties for C3 herbs, though available measurements are limited to a narrow [$O_3$] range of 70–95 ppbv (Supplementary Data 1). More observations with a wide range of [$O_3$] are required to derive a robust GPP response to $O_3$.

GPP increases with photosynthetically active radiation (PAR) but at different rates for diffuse and direct light. Analyses at 24 FLUXNET sites (Supplementary Fig. 1) show that GPP grows faster with light when light is more diffusive (Supplementary Fig. 2). The YIBs model generally reproduces GPP responses to both diffuse and direct light. Observed GPP increases by 0.3–0.7% for 1 W m$^{-2}$ enhancement in diffuse PAR (Fig. 2). The modeled GPP sensitivities are close to observations for tree species but show large uncertainties for grass and crops. These sensitivities are derived assuming no changes in direct light or other meteorological variables. Appearance of aerosols (or cloud) will increase diffuse radiation but decrease direct radiation. The reduction of direct light can offset the benefit of diffuse fertilization, leading to positive GPP responses at low–medium diffuse fraction but negative responses at high diffuse fraction[16,17].

### Thresholds for cloud diffuse effects on GPP.
Site-level studies reveal that plant photosynthesis reaches the maximum when diffuse fraction is between 0.4 and 0.6, no matter for cloud or aerosols[16,17,26]. Theoretically, each location on the Earth has a specific cloud amount leading to optimized GPP. It is important to estimate such threshold so as to quantify potentials for aerosol DFE. Here, we perform 20 sensitivity experiments with cloud scaling factors from 0 to 10 without aerosol radiative effects (Methods). Cloud feedbacks to climate (e.g., cooling and rainfall) are not considered due to the large uncertainties. Simulation results are compared with identify the maximum GPP and derive the corresponding cloud scaling factor (defined as cloud scaling threshold). The model predicts maximum global GPP at cloud scaling of 0.9 (Supplementary Fig. 3), suggesting that ecosystem productivity is in general almost optimized at the current cloud amount. Regionally, cloud scaling thresholds are usually <1 for boreal and tropical forest (Fig. 3), where cloud cover is too dense (Supplementary Fig. 4) and limits plant photosynthesis. In contrast, the cloud scaling threshold is >1 over arid and semi-arid areas (Fig. 3), suggesting that light availability is not saturated with present-day cloud amount over those regions and increasing atmospheric aerosol loading has the potential to further increase plant photosynthesis.

### Fire-induced perturbations in $O_3$ and aerosols.
Global fire emissions cause large enhancements in surface [$O_3$] and AOD (Supplementary Fig. 5). The largest enhancement of [$O_3$] (grid maximum >15 ppbv) is found in southern Africa (Supplementary Fig. 5a), where fires on average contribute 6.9 ppbv (21.3%) to the local [$O_3$]. Furthermore, fires increase [$O_3$] by 2.9 ppbv (12.0%) in central South America, 3.1 ppbv (14.8%) in Indonesia, 2.1 ppbv (6.4%) in boreal Asia, and 1.9 ppbv (5.8%) in boreal North America on an annual mean basis. Regional perturbations are stronger in the corresponding fire seasons. For example, fires increase [$O_3$] by 4.4 ppbv in eastern Siberia and 5.9 ppbv in Alaska during boreal summer. Globally, fires enhance surface $O_3$ by at least 0.5 ppbv over 2/3 land grids, for which we find a negative correlation of $-0.36$ ($p < 0.01$) between ambient [$O_3$] (all emissions but fires, Supplementary Fig. 5c) and fire [$O_3$] (Supplementary Fig. 5a), suggesting that most fire activity occurs in wildlands or less developed regions where anthropogenic emissions are usually low. In addition to $O_3$, fires contribute to regional AOD by 39.2% in southern Africa, 33.0% in tropical Amazon, and 31.8% in Indonesia on an annual mean basis (Supplementary Fig. 5b). At the global scale, fires promote [$O_3$] by $5.4 \pm 0.6\%$ and AOD by $9.6 \pm 1.9\%$ over continents on the decadal average during 2002–2011, with the maximum in the year 2003.

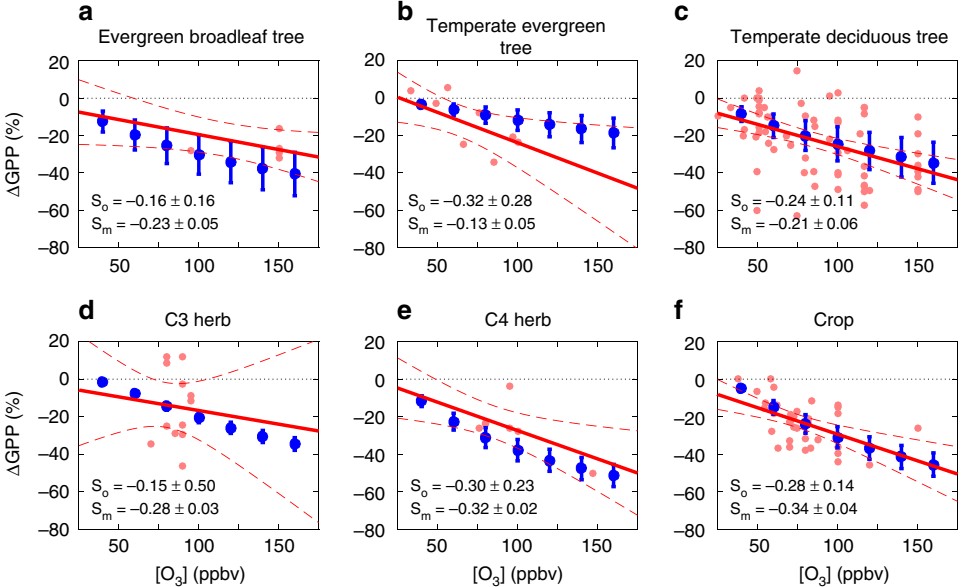

**Fig. 1** Observed and predicted GPP changes by $O_3$. Results shown are the percentage changes in gross primary productivity (GPP) for six main plant functional types (PFTs). Red points on each panel represent results summarized from literature (Supplementary Data 1). The linear regression is denoted as a red solid line, with 95% confidence intervals shown as dashed lines. Blue points represent simulated GPP changes from offline sensitivity experiments (Methods), with error bars indicating the range of prediction from low to high $O_3$ damaging sensitivities. The slopes of observed ($S_o$, mean ± 95% confidence interval) and modeled ($S_m$, mean ± (high-low)/2 sensitivity) GPP–$O_3$ sensitivity is shown on each panel

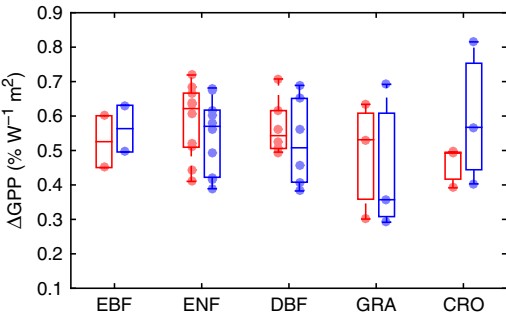

**Fig. 2** Observed and predicted GPP responses to diffuse radiation. Results shown are the percentage changes in GPP for five PFTs due to an increase of 1 W m$^{-2}$ diffuse radiation. The changes of GPP are calculated for 24 FLUXNET sites (Supplementary Fig. 1) using both observed (red) and simulated (blue) data. These sites are composed of evergreen broadleaf forest (ENF), evergreen needleleaf forest (ENF), deciduous broadleaf forest (DBF), C3 grassland (GRA), and cropland (CRO). At each site, half-hourly GPP is grouped at specific PAR intervals for all the high diffusive moments (diffuse fraction > 0.8). The derived slopes of GPP–PAR relationships (blue for observations and green for simulations in Supplementary Fig. 2) are shown above. Each point represents the slope for one site, with the boxplot showing the median values and related ranges for a PFT

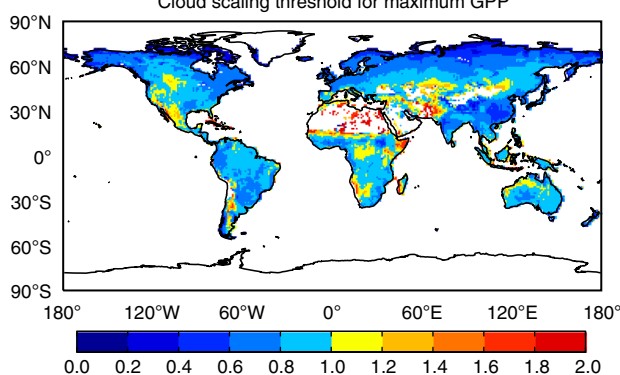

**Fig. 3** Cloud scaling threshold for maximum GPP. In total, 20 simulations are performed by scaling daily cloud amount with factors ranging from 0 to 10 (Methods). The derived diffuse and direct radiation from each scaling are then used as input for the vegetation model to estimate GPP at varied cloud scalings. For each grid, the scaling factor leading to maximum GPP is derived and shown. Blue patches indicate that maximum GPP appears if local scaling factor < 1, suggesting that current cloud amount is higher than the optimal level for plant photosynthesis. Red/yellow patches indicate that maximum GPP emerges if cloud amount is higher than current level, suggesting that aerosol radiative effects have potentials to increase GPP at those grids

**GPP reductions by $O_3$.** Plants can have low or high sensitivity to $O_3$ damage due to their varied tolerance[25]. We calculate the ozone damage as the central value of the low and high sensitivity impact results, and evaluate uncertainties as the range from low to high sensitivity. With all emissions except from fires, global GPP is reduced by 4.0 ± 1.9 Pg C yr$^{-1}$ (2.9 ± 1.4%, mean ± uncertainties) due to $O_3$ damage losses compared with $O_3$-free GPP (Fig. 4a). Such damaging effects are more serious in eastern US (−8.2%), Europe (−4.7%), and eastern China (−7.5%), where anthropogenic emissions are large. The $O_3$ damages are likely overestimated in eastern US because of the high biases in modeled

[$O_3$] (Methods). Inclusion of fire emissions causes additional $O_3$ damage of 0.92 ± 0.44 Pg C yr$^{-1}$ globally (Fig. 4b).

Large fire $\Delta[O3]$ does not always cause high GPP reductions, and vice versa. Over regions with fire-induced $\Delta[O_3]$ > 1 ppbv (Supplementary Fig. 5a), $O_3$ enhancements cause additional GPP reductions of 1.7% in central Africa, 1.5% in Indonesia, 0.5% in central South America, and 0.6% in northeastern Asia during 2002–2011. However, fire $O_3$ causes limited impacts over southern Africa, Australia, and boreal North America. We find that the average stomatal conductance over fire-$O_3$-damaging

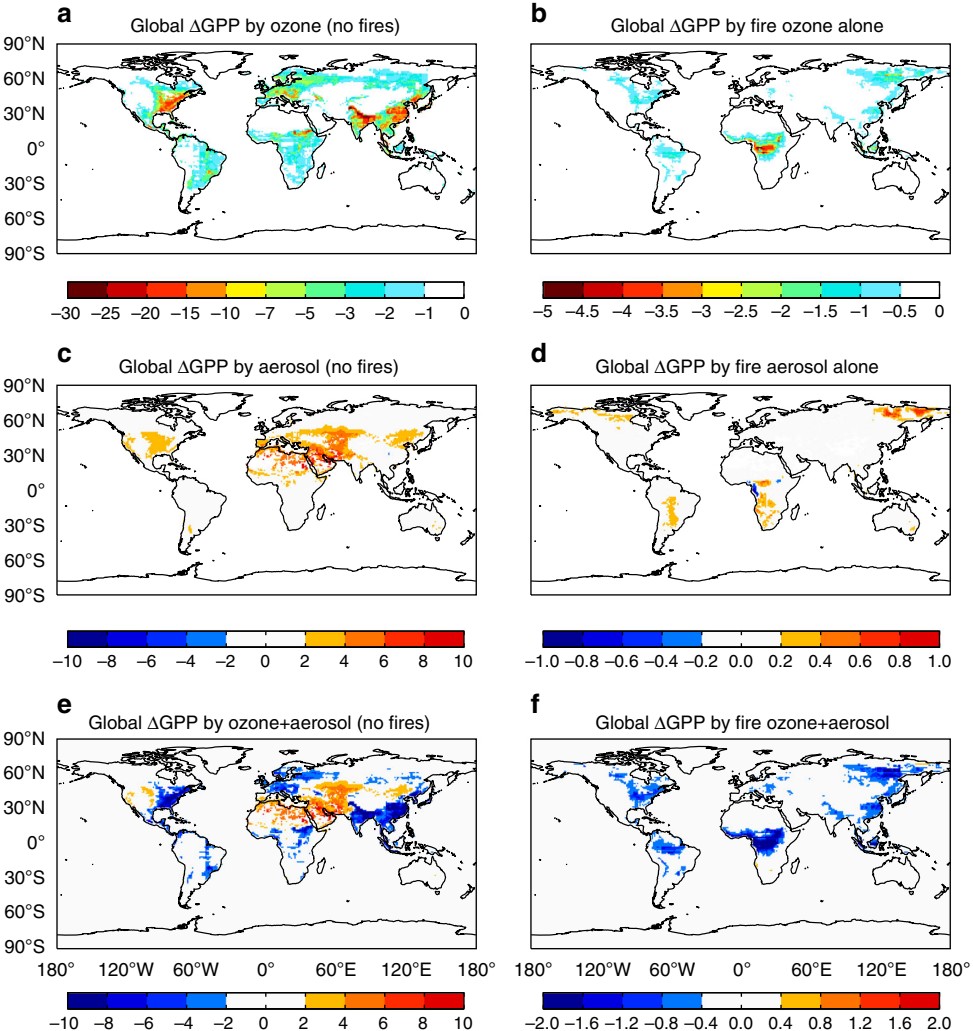

**Fig. 4** Spatial pattern of GPP changes due to ozone and aerosols. Results shown are the percentage changes in GPP caused by **a**, **b** surface ozone, **c**, **d** aerosols, and **e**, **f** their combined effects. The effects of air pollution from all sources (anthropogenic + natural) except fire emissions are shown in the left, and those from fire emissions alone are in the right

areas (high fire $O_3$ with high damages, red patches in Supplementary Fig. 6) is 79.2 mmol m$^{-2}$ s$^{-1}$, much higher than the average of 19.0 mmol m$^{-2}$ s$^{-1}$ over fire-$O_3$-resilient regions (high fire $O_3$ but with limited damage, green patches in Supplementary Fig. 6), suggesting that the higher stomatal conductance drives the larger $O_3$ uptake. On the other hand, we find additional GPP reductions of 0.6% in eastern US and 0.5% in eastern China (Fig. 4b), where the fire-induced $O_3$ enhancement is usually less than 1ppbv (Supplementary Fig. 5a). Over those regions (blue patches in Supplementary Fig. 6), the average ambient [$O_3$] (all sources except fire emissions, Supplementary Fig. 5c) is 34.2 ppbv and stomatal conductance is 75.1 mmol m$^{-2}$ s$^{-1}$, both of which are higher than the corresponding values of 28.0 ppbv and 51.7 mmol m$^{-2}$ s$^{-1}$ over fire-prone regions (red plus green patches in Supplementary Fig. 6). As a result, the high background [$O_3$] and stomatal conductance together provide such a sensitive environment that even a mild increase in [$O_3$] by fires may cause a discernable influence.

**GPP changes by aerosol DFE**. In contrast to the $O_3$ impacts, high AOD does not always result in large responses in GPP (Fig. 4c). Moderate AOD promotes plant photosynthesis by increasing

diffuse radiation, while dense aerosol loading plays the opposite role due to the strong attenuation of direct light[16]. In addition, aerosol DFE is highly sensitive to cloud amount which may have saturated the supply of diffuse radiation at the cost of total light availability[17,27]. Sensitivity tests show that aerosols are beneficial to plant photosynthesis with zero or low cloud coverage, but cause limited or even negative GPP responses when cloud fraction is high[28,29]. Our simulations with zero cloud coverage show widespread GPP enhancement of 6–30% over subtropical continents (Supplementary Fig. 7a), especially in China where dense aerosols significantly increase regional diffuse fraction (Supplementary Fig. 8a). However, with the inclusion of cloud effects, regions with ΔGPP > 2% are limited to western Asia, North Africa, and central US (Fig. 4c), most of which are arid or semi-arid regions with average cloud fraction of 39.4% (Supplementary Fig. 4). For these regions, the cloud scaling threshold is higher than 0.8 (Fig. 3) so that local cloud is not too abundant to inhibit aerosol DFE. For other land regions where the cloud scaling threshold is lower than 0.8 (Fig. 3), the dense cloud masks aerosol DFE and leaves very limited ΔGPP (Fig. 4c). The average GPP is 1.2 g C m$^{-2}$ day$^{-1}$ over the arid/semi-arid regions with ΔGPP > 2%, much lower than the average of 3.5 g C m$^{-2}$ day$^{-1}$ in the wet regions with ΔGPP < 2%, leading to a limited potential for aerosol DFE on the continental scale. Globally, aerosols from non-fire

sources promote annual GPP by 1.0 Pg C yr$^{-1}$ (0.8%), much lower than the enhancement of 8.1 Pg C yr$^{-1}$ under clear-sky conditions.

Fire aerosols alone cause limited impacts on plant photosynthesis except over central and southern Africa, central South America, and eastern Siberia ($|\Delta GPP| > 0.2\%$, Fig. 4d), where fire aerosols enhance diffuse fraction by 1.4%, 0.7%, and 0.5% (Supplementary Fig. 8d). We find contrasting responses of GPP to fire aerosols in Africa, with negative changes along the western coast while positive responses occur in inner areas. Such discrepancy is related to the seasonality of biomass burning and cloud fraction. African fire pollution is confined between 0–15ºS in boreal summer (June–August) and shifts northward to 0–8ºN in winter (December–February). Meanwhile, dense cloud overlaps the fire domain (especially over the western coast) in winter but shifts northward to an offset location with fires in the summer (Supplementary Fig. 4). On average, the summertime cloud fraction is 72.0% over the coastal regions with negative aerosol effects, but is much lower at 48.5% over the inner areas where $\Delta GPP > 0.2\%$. The dense clouds limit DFE of fire aerosols along the coast due to the attenuation of total available sunlight. The largest GPP percentage change of 1% is found in eastern Siberia, where diffuse fraction is increased only 0.5% by fire aerosols (Supplementary Fig. 8d). Sensitivity tests without cloud effects also show large responses in GPP at high latitudes (Supplementary Fig. 7b), where the changes in diffuse fraction are smaller than those in tropical regions (Supplementary Fig. 8b). Plants at high latitudes have much larger shaded portion than that at low latitudes because of the smaller solar zenith in high latitudes (Methods). As a result, the boreal forest is more sensitive than tropical trees to the same fractional change in diffuse radiation. On the global scale, fire aerosols enhance GPP by 0.05 Pg C yr$^{-1}$, lower than the value of 0.7 Pg C yr$^{-1}$ under clear-sky conditions (Supplementary Fig. 7b).

**Net effects of O$_3$ and aerosols to GPP.** Combined aerosol and O$_3$ impacts on GPP show offset spatial patterns globally (Fig. 4e). Aerosol radiation fertilization effects are more dominant over arid and semi-arid areas, where human activity is usually limited. In contrast, O$_3$ damage effects are prominent over regions with high population density, where both anthropogenic emissions and vegetation coverage are high. The net impact of fire pollution on GPP is dominated by the O$_3$ effect (Fig. 4f), though both O$_3$ and aerosols are generated by the same fires. On the global scale, we quantify that fire emissions indirectly reduce GPP by 0.86 Pg C yr$^{-1}$ (0.6%) during 2002–2011, resulting from a reduction of 0.91 Pg C yr$^{-1}$ by O$_3$ and an increase of 0.05 Pg C yr$^{-1}$ by aerosols (Fig. 5a). The fertilization effect of fire aerosols is very limited, likely because fire emissions predominantly occur in tropical forests where dense cloud masks the aerosol effects. The overlap of tropical forests and cloud typifies the intimacy of biosphere–atmosphere coupling: the vegetation grows in regions with abundant precipitation while strong evapotranspiration generates more cloud and consequent rainfall[30].

Global fire emissions show interannual variations even with regional offsets[1]. The largest increase of fire O$_3$ of 2.1 ppbv occurs in the year 2003, leading to the highest GPP reduction of 0.99 Pg C yr$^{-1}$ globally (Fig. 5c). In contrast, the smallest GPP reduction of 0.85 Pg C yr$^{-1}$ is found in the year 2011, when the average fire-induced [O$_3$] is 1.9 ppbv globally. On average, the interannual variation (0.05 Pg C yr$^{-1}$, one standard deviation for 2002–2011) accounts for 5% of the mean $\Delta GPP$ by fire pollution, much smaller than the uncertainties of 43% driven by O$_3$ damage sensitivity (0.37 Pg C yr$^{-1}$, half of the range from low to high sensitivities). Regionally, fire perturbations cause large year-to-

year variability (Fig. 5d). For example, $\Delta GPP$ ranges from −0.9% (2005) to −0.2% (2011) in the Amazon, −1.5% (2002) to −0.2% (2004) in boreal Asia, and −3.6% (2006) to −0.5% (2008) in Indonesia. In contrast, changes in GPP are relatively stable in southern Africa (−2% in 2003 to −1.5% in 2011), because most of the fires there are caused by human activities with small interannual variability[1,31].

**Discussion**
Compared with simulations by Pacifico et al.[21], which estimated that fire O$_3$ reduces GPP by 230 Tg C yr$^{-1}$ in Amazon forest, this study predicts a lower O$_3$ damage of 137 Tg C yr$^{-1}$ over the same region (Fig. 4b). Our predicted GPP reduction is smaller, likely because Pacifico et al.[21] overestimated background [O$_3$] by 5–15 ppbv in Amazonia. In our simulations, for some regions and seasons, the low ambient [O$_3$] cannot induce substantial damage even combining O$_3$ from biomass burning. For fire aerosols, Rap et al.[22] estimated that DFE enhances the NPP of Amazon Basin by 78–156 Tg C yr$^{-1}$. Over the same domain, offline simulations in this study show a GPP increase of only 14 Tg C yr$^{-1}$ due to the DFE of fire aerosols (Fig. 4d). The GEOS-Chem simulation with satellite-based emission inventories may underestimate fire AOD compared with MODIS[32]. However, sensitivity simulations with doubled or tripled fire aerosols does not increase Amazon GPP substantially (Supplementary Fig. 9). The average cloud amount in this region is 69% on an annual mean basis and remains as high as 50% during the dry season (Supplementary Fig. 4). A previous study reported decreases in plant photosynthesis when cloud cover is higher than 48%[17]. The dense cloud over Amazonia implies a low potential for increasing diffuse radiation, and as a result limits DFE of aerosols from fires. On the other hand, we predict relatively high responses (0.2–0.4%) in GPP south of Amazonia (Supplementary Fig. 9), where cloud fraction is 44% during the dry season.

Derived fire pollution effects on ecosystem productivity are influenced by uncertainties in O$_3$ damage sensitivity, cloud amount, and aerosol properties. For O$_3$ damage, applying low to high sensitivity parameters results in varied $\Delta GPP$ from −1.29 to −0.55 Pg C yr$^{-1}$ (Fig. 5b). Predicted GPP reduction by fire O$_3$ is high over tropical Africa (Fig. 4b), where the dominant PFT is evergreen broadleaf forest (EBF). The O$_3$ damage scheme proposed by Sitch et al.[25] was developed based on temperate plants. Compared with limited measurements, the derived GPP sensitivity is larger by 37.5% for EBF (Fig. 1). A recent meta-analysis[33] shows that the plant O$_3$ sensitivity is closely related to leaf mass per area (LMA), suggesting that tropical trees may have higher resistance to ozone damage due to their larger LMA (average 107–121 g m$^{-2}$)[34] than temperate trees (usually < 100 g m$^{-2}$)[33]. As a check, we reduce the O$_3$ sensitivity coefficients of EBF in the YIBs model by 33% while holding unchanged the coefficients for other PFTs (Methods). The new simulations show that global GPP is reduced 3.7 ± 1.8 Pg C yr$^{-1}$ by fire-free O$_3$ and 0.81 ± 0.40 Pg C yr$^{-1}$ by fire O$_3$. These values are slightly smaller than the results of 4.0 ± 1.9 Pg C yr$^{-1}$ and 0.92 ± 0.44 Pg C yr$^{-1}$ calculated with original sensitivity coefficients, but do still mean that fire O$_3$ damage dominates over fire aerosol effects at the global scale.

For aerosol effects, simulated aerosol DFE is sensitive to cloud properties. In the current runs, daily cloud fraction and liquid water path is applied though cloud property may have distinct spatiotemporal variations. As a check, we use 3-hour cloud variables from CERES to update fire aerosol DFE (Methods). Simulated $\Delta GPP$ by fire aerosols with 3-hour cloud is 0.01 Pg C yr$^{-1}$, slightly lower than the value of 0.05 Pg C yr$^{-1}$ obtained using daily cloud (Fig. 5b). Current simulations employ monthly fire emissions, which may lead to inconsistency between fire

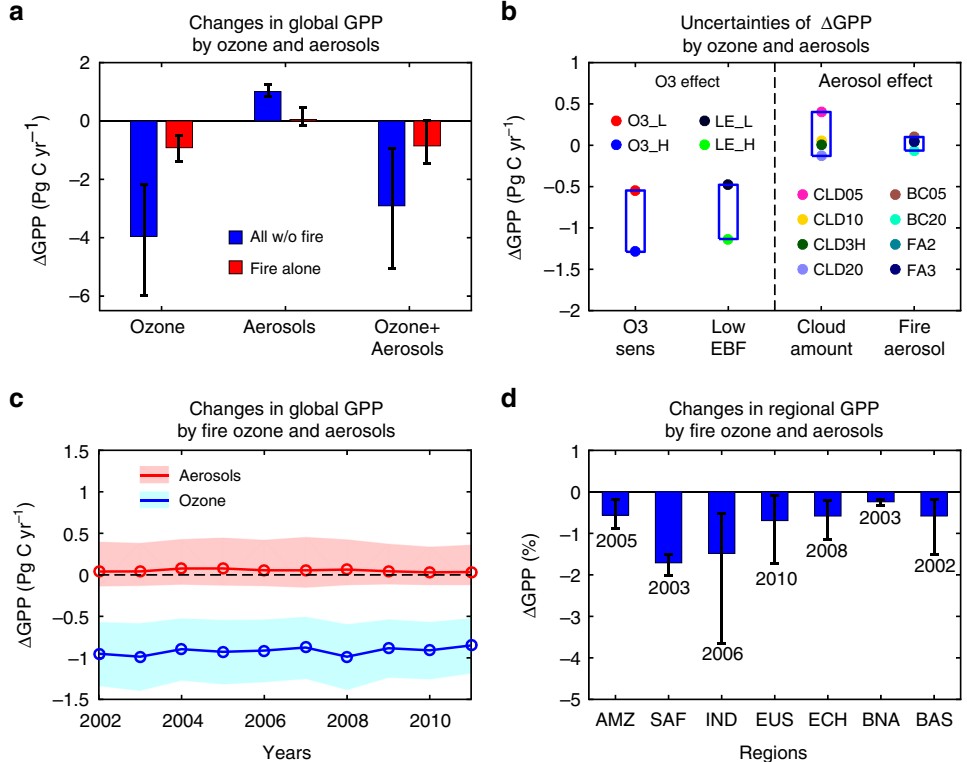

**Fig. 5** Changes in GPP caused by fire ozone and aerosols. **a** The changes caused by air pollution from all sources (anthropogenic + natural) except fire emissions are shown in blue and those from fire emissions alone are shown in red. The errorbar of $O_3$ represents low to high damaging sensitivity. The errorbar of aerosols indicates uncertainties from 0.5 to 2.0 cloud scalings. **b** Uncertainties for ozone effects mainly originate from damaging sensitivity and relatively lower responses of evergreen broadleaf forest, while uncertainties for aerosol effects are mainly from cloud amount and perturbations in fire aerosols (e.g., content of black carbon, or amount of total fire particles, see Supplementary Table 8). **c** Global GPP changes due to fire ozone (blue) and aerosols (red) show moderate interannual variations. Blue shadings represent uncertainties due to ozone damaging sensitivity. Red shadings indicate uncertainties of aerosol effects caused by cloud scalings from 0.5 to 2.0. **d** Fire pollution causes distinct responses to regional GPP (see Supplementary Fig. 5 for region names), with the maximum perturbations in southern Africa (SAF), the minimum in boreal North America (BNA), and the largest interannual variation in Indonesia (IND). Error bars indicate ranges of GPP changes from low to high years. The years leading to maximum regional GPP changes are denoted

events and cloud status. Fires tend to occur in dry periods when the cloud amount is relatively low, and fire aerosols can inhibit cloud and precipitation through fire-meteorology interactions[35]. To place bounds on the DFE uncertainty due to clouds, we perform two sensitivity experiments that alter the cloud fraction and liquid water path. First, we reduce the observed cloud fraction and liquid water path by 50% (Methods). In this case, fire aerosols cause a stronger fertilization effect of 0.40 Pg C $yr^{-1}$ to global GPP. At the same time, fire aerosols themselves can increase cloud condensation nuclei and consequently increase cloud depth and amount[36]. Therefore, a second sensitivity experiment is performed with doubled observed cloud fraction and liquid water path. In this case, fire aerosols result in a negative GPP change of $-0.13$ Pg C $yr^{-1}$ (Fig. 5b).

In addition to cloud properties, uncertainties in aerosol radiative effects can affect GPP responses. Since monthly fire pollution may underestimate high density of particles during episodes, we perform two simulations with doubled and tripled fire aerosols (Methods). Predicted ΔGPP in these two simulations are respectively, 0.06 Pg C $yr^{-1}$ and 0.04 Pg C $yr^{-1}$, very similar to the value of 0.05 Pg C $yr^{-1}$ with the original fire emissions. Fire aerosols are dominated by organic (OC) and black carbon (BC), which have distinct optical properties. Here, two sensitivity experiments are performed that alter the BC loadings and BC/OC ratios. Simulation BC05 assumes that the fraction of BC in the fire pollution is reduced by 50% while OC increases by 5%, because the emission

factor of BC is ~1/10 of that for OC[37] in fires. Simulation BC20 assumes the loading of BC is doubled while OC decreases by 10%. Predicted diffuse fraction is similar between BC05 and BC20, but the total PAR is smaller in BC20 because BC is more absorptive than OC. Consequently, ΔGPP by fire aerosols in BC05 is 0.10 Pg C $yr^{-1}$, higher than the value of $-0.06$ Pg C $yr^{-1}$ in BC20. In summary, uncertainties in composition and optics of fire aerosols cause varied GPP responses from $-0.06$ to 0.10 Pg C $yr^{-1}$, generally smaller than those of $-0.13$ to 0.40 Pg C $yr^{-1}$ due to changed cloud properties (Fig. 5b). Finally, aerosols alter meteorology (e.g., temperature and precipitation) and cloud (both semi-direct and indirect effects), which in turn affect plant photosynthesis and carbon assimilation[19,20,38]. We do not include these feedbacks due to the large uncertainties. Furthermore, deposition of fire aerosols may cause physiological (e.g., altering stomatal functions[39]) and biogeochemical (e.g., nitrogen deposition effects[40]) responses but these potential impacts are ignored in the current study.

This study reveals an indirect carbon loss due to the combined effects of $O_3$ stomatal uptake and aerosol diffuse fertilization by fire emissions. The damage to ecosystem productivity not only occurs in the fire regimes, but also over the downwind areas through long-range transport of air pollution. The net impact is estimated to be a 0.6% reduction per year in global GPP by fire pollution. In comparison, drought is estimated to reduce net primary production by 0.55 Pg C (1%) during 2000–2009[41]. The

fire pollution-induced productivity inhibition (0.6% per year) is considerably stronger than the drought effect (0.1% per year) for the same period. Regionally, fire pollution causes GPP losses of up to 3.6% during some large fire years. Over tropical Africa, fire pollution induces stable GPP reductions of 1.5–2% every year. Such perturbations may result in the loss of land carbon storage and exacerbate the global warming trend due to increasing the atmospheric $CO_2$ burden. Although some studies[42] suggest that increased temperature is beneficial to the biosphere by enhancing GPP at the rate of 0.8–0.9 Pg C $yr^{-1}$ $°C^{-1}$, the associated increase in fire activity[43,44] as well as background $O_3$[45] may weaken the global and regional GPP response to a warming world.

## Methods

**Models.** We apply a suite of models in offline mode for this study (Supplementary Fig. 10). The chemical transport model (CTM) GEOS-Chem[23] (version 9-1-2) is used to predict fire-induced changes in AOD and surface $O_3$. The GEOS-Chem model simulates gas-phase chemistry ($NO_x$, $HO_x$, $O_x$, CO, $CH_4$, and non-methane volatile organic compounds (NMVOCs)) and aerosols (sulfate, nitrate, black (BC) and organic carbon (OC), dust, sea salt), and their interactions[46,47]. The CTM is driven with 3-D meteorology from NASA's Modern-Era Retrospective Analysis for Research and Applications (MERRA)[48] and 2-D surface emissions from both anthropogenic and natural (e.g., soil $NO_x$, biogenic VOC) sources. Wildfire emissions of $SO_2$, $NO_x$, CO, NMVOCs, BC, and OC are adopted from Global Fire Emission Database (GFED)[1] version 3.

The CRM simulates aerosol-induced perturbations in surface radiative fluxes. The model is the standalone version of the radiation module used by the NCAR Community Climate Model (http://www.cesm.ucar.edu/models/). It calculates aerosol scattering and absorption within 20 vertical layers at hourly time interval. Wavelength- and species-dependent optical parameters, including extinction coefficient, single-scattering albedo, and asymmetry parameter, are adopted from multiple sources with reasonable calibrations[28]. The CRM is driven with hourly 3-D meteorology (e.g., temperature and humidity) from MERRA reanalyses and daily cloud profile (e.g., cloud fraction and liquid water path) from CERES SYN1deg assimilation data product (http://ceres.larc.nasa.gov), which is developed based on satellite retrievals from the Moderate Resolution Imaging Spectroradiometer (MODIS) and the Visible and Infrared Sounder (VIRS).

YIBs is a process-based vegetation model that simulates the global carbon cycle with dynamical prediction of leaf area index (LAI) and tree height[24]. The model considers eight PFTs, including evergreen needleleaf forest, deciduous broadleaf forest, EBF, shrubland, tundra, C3/C4 grass, and C3 crops. The satellite-based land types and cover fraction[49] are aggregated into these eight PFTs and used as input. The model is driven with hourly 2-D meteorology (e.g., surface air temperature, specific humidity) and 3-D soil variables (e.g., soil temperature and moisture) from WFDEI[50] (WATCH Forcing Data methodology applied to ERA-Interim data). Leaf-level photosynthesis is simulated with the well-established Farquhar et al.[51] scheme and is coupled to stomatal conductance with the Ball–Berry scheme[52]. The canopy is divided into multiple layers with separation of diffuse and direct light components on sunlit and shaded leaves[53]. Light intensity decreases exponentially with LAI and becomes more diffusive when penetrating into the deep canopy. At each layer, plant photosynthesis is calculated as:

$$A = f_{sl} \cdot A_{sl} + (1 - f_{sl}) \cdot A_{sh} \qquad (1)$$

where $A_{sl}$ and $A_{sh}$ are the photosynthetic rates of sunlit and shaded leaves, respectively. The sunlit leaves receive both direct and diffuse light while the shaded ones receive only diffuse radiation. The areal fraction of sunlit leaf $f_{sl}$ is calculated as:

$$f_{sl} = e^{-\sigma L} \qquad (2)$$

where $L$ is the LAI at a certain canopy layer. The extinction coefficient $\sigma = 0.5/\cos\alpha$ is dependent on solar zenith $\alpha$, and will be larger at higher latitudes, leading to smaller (larger) fraction of sunlit (shading) leaves.

The YIBs model calculates $O_3$ damage to photosynthesis following the Sitch et al.[25] flux-based scheme, which assumes GPP reduction rate $F$ as a function of stomatal $O_3$ flux $F_{O3}$:

$$F = \begin{cases} -a \cdot (F_{O3} - T_{O3}), & \text{if } F_{O3} > T_{O3} \\ 0, & \text{if } F_{O3} \leq T_{O3} \end{cases} \qquad (3)$$

Here, $a$ is the damaging sensitivity coefficient and $T_{O3}$ is the flux threshold. The coefficient $a$ varies among PFTs and shows low to high sensitivities for species within the same PFT[25]. The coefficient $a$ can be calibrated using observed GPP–$O_3$ responses (Fig. 1). The $F$-value becomes negative only if $F_{O3}$ is higher than $T_{O3}$, the latter of which is an indicator of $O_3$ tolerance for plants. Stomatal $O_3$ flux $F_{O3}$ is dependent on both stomatal resistance and ambient $[O_3]$:

$$F_{O3} = \frac{[O_3]}{r_b + k \cdot r_s} \qquad (4)$$

where $[O_3]$ is $O_3$ concentration at top of the canopy, $r_b$ is the boundary layer resistance, and $r_s$ is the stomatal resistance ($=1/g_s$, $g_s$ is stomatal conductance). In general, high $[O_3]$ and high stomatal conductance ($=$low $r_s$) results in high $F_{O3}$ and high consequential damage. On the other hand, high $[O_3]$ does not cause damages to photosynthesis if stomatal conductance is too low to accumulate $F_{O3}$ up to the damaging threshold $T_{O3}$.

**Simulations and validation.** We perform two GEOS-Chem runs, three CRM runs, and eleven YIBs runs to isolate ozone and aerosol impacts on GPP from fire and non-fire sources (Supplementary Table 1). The GEOS-Chem runs, GC_NOFIRE and GC_FIRE, are driven with the same meteorology and emissions, except that the former omits fire emissions. The CRM runs calculate radiative fluxes based on aerosol profiles simulated by GESO-Chem, with no aerosol effects in CRM_NA, effects of non-fire aerosols in CRM_AA, and effects of all (non-fire plus fire) aerosols in CRM_FA. The YIBs runs separate and/or combine individual air pollution impacts on biosphere. For example, three runs, YIBS_NA, YIBS_AA, and YIBS_FA, neglect ozone effects but are driven with PAR from aerosol-free, non-fire aerosol, and all aerosol effects, respectively. In contrast, four YIBs runs, YIBS_NOH, YIBS_NOL, YIBS_FOH, and YIBS_FOL, omit aerosol effects but considers ozone effects from non-fire or all sources with either low or high damaging sensitivities. Finally, the other four YIBs runs, YIBS_AAOH, YIBS_AAOL, YIBS_FAOH, and YIBS_FAOL, consider both aerosol and ozone effects but differ from each other depending on whether fire emissions are included and which level of ozone damaging is applied. All simulations are performed for 2001–2011 and the last 10 years are used for analyses. Simulated aerosol profiles with coarser resolution (5° longitude by 4° latitude) from GEOS-Chem are conservatively downscaled to 1.3° × 1° for CRM, which calculates aerosol-induced changes in diffuse and direct PAR based on Mie scattering processes. Simulated $O_3$ from GEOS-Chem and radiation from CRM are further regridded onto 1° × 1° for YIBS model to calculate GPP responses to $O_3$ damaging and aerosol DFE (Supplementary Fig. 9).

The YIBs model simulates reasonable GPP spatial pattern compared with a benchmark product upscaled from FLUXNET site data (Supplementary Fig. 11). The simulation matches benchmark well with high correlation coefficient of 0.9 ($p < 0.01$) and low relative bias of 3.6%, except that simulated GPP is higher by 37.6% in Amazon and 44.1% in tropical Africa. However, site-level simulations at eight EBF sites from FLUXNET network (Supplementary Fig. 12) show that the model actually overestimates GPP by 9–22% in Amazon (BR-Sa3 and GF-Guy) and 19% in Africa (GH-Ank). The YIBs also predicts reasonable GPP responses to $O_3$ and diffuse radiation. To test ozone damaging scheme, we perform 17 sensitivity tests (Supplementary Table 2) with different levels of $[O_3]$ and sensitivity coefficients. We compare the $O_3$-affected GPP with the $O_3$-free GPP from the baseline simulation (YIBS_O000) to derive the damaging percentages to GPP, which are compared with values for different PFTs from 60 published literatures (Supplementary Data 1) shown in Fig. 1. Additional sensitivity experiments are performed with $O_3$ sensitivity coefficients of EBF lower by 33% (Supplementary Table 3), because a recent meta-analysis shows that tropical trees may have lower sensitivity to ozone damage due to their larger LMA[33]. For diffuse effect, we perform simulations at FLUXNET sites providing long-term (>3 years) hourly diffuse radiation. In total, 24 sites are selected with wide geographic distribution (Supplementary Fig. 1). For these sites, we separate hours dominated by diffuse (diffuse fraction > 0.8) and direct (diffuse fraction < 0.2) radiation, and then aggregate both observed and simulated GPP into discrete bins based on PAR intervals. In this way, we isolate GPP responses to diffuse/direct radiation by smoothing influences of other meteorological forcings (Supplementary Fig. 2).

Sensitivity experiments are performed to estimate GPP uncertainties due to cloud and aerosol properties. To derive cloud thresholds for optimized GPP, we perform 20 sets of simulations with cloud scaling factors ranging from 0 to 10 (Supplementary Tables 4, 5). In this way, 3-D cloud profile is retained and cloud fraction in multiple layers can be 100%. Predicted total PAR in these runs always decrease with increased scaling factors (Supplementary Table 4). However, diffuse fraction increases at beginning but decreases with large cloud amount, because diffuse light reduces after the scaling of 1.5 (Supplementary Fig. 3). To estimate the uncertainties of aerosol DFE, another 12 sets of runs are performed (Supplementary Tables 6, 7) by assuming doubled cloud amount (C20), half cloud amount (C05), 3-hour cloud variables (C3H), no cloud (C00), doubled fire BC (BC20), half fire BC (BC05), and doubled (FA2) and tripled (FA3) total fire aerosols. Results from these sensitivity experiments are shown in the main text with instructions listed in Supplementary Table 8.

Simulated AOD based on aerosols from GEOS-Chem model (Supplementary Fig. 13a) reproduces the observed (Supplementary Fig. 13c) spatial distribution retrieved by satellites. The model overestimates AOD by 25.1% in eastern China, but underestimates AOD by 34.7% in Amazonia, 44.6% in India and 58.0% in western US. The satellite data show no values over North Africa, where the model predicts high AOD due to dust, because the bright surface results in the failure of retrieval. On the global scale, simulated AOD correlates with observations at $R =$

0.63 ($p < 0.01$) with a relative bias of 5.1% (Supplementary Fig. 13e). Observations of surface [$O_3$] are very limited. Here, we combine long-term monitoring data from US and Europe[54] with records at 1580 sites from Chinese Ministry of Ecology and Environment (http://english.mep.gov.cn/, Supplementary Fig. 13d). The comparison shows that GEOS-Chem simulation (Supplementary Fig. 13b) generally overestimates surface $O_3$, leading to a positive bias of ~20% on the global scale (Supplementary Fig. 13f). With simulated aerosols from GEOS-Chem, the CRM reproduces surface solar insolation with biases of only 1.8% for all-sky and 2.6% for clear-sky conditions (Supplementary Fig. 14). Simulated clear-sky aerosol direct radiative effect (DRE) shows reasonable seasonality compared with other models summarized in Yu et al.[55] (Supplementary Fig. 15). On the annual mean basis, the model predicts aerosol DRE of $-6.3$ W m$^{-2}$ at surface and $-3.1$ W m$^{-2}$ at top of atmosphere, close to the averages of $-7.6$ and $-3.0$ W m$^{-2}$ from an ensemble of models.

**Code availability**. The YIBs model was developed by Xu Yue and Nadine Unger with code sharing at https://github.com/YIBS01/YIBS_site. The CRM model was developed by Prof. Charles Zender at University of California, Irvine (https://www.ess.uci.edu/~zender/). The GEOS-Chem model was developed by the Atmospheric Chemistry Modeling Group at Harvard University led by Prof. Daniel Jacob (http://acmg.seas.harvard.edu/geos/).

## Data availability

Results of all simulations are available upon request, including 64 sets of YIBs simulations, 35 sets of CRM simulations, and 2 sets of GEOS-Chem simulations. Site-level eddy covariance flux data from FLUXNET is available online (http://fluxnet.fluxdata.org/). Data of the meta-analysis of $O_3$ damage to photosynthesis (Fig. 1) are listed in Supplementary Data 1.

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

## Acknowledgements

The authors would like to thank Dr. Danica Lombardozzi for providing the meta-analysis measurement data of $O_3$ vegetation damage. Xu Yue acknowledges funding support from the Strategic Priority Research Program of the Chinese Academy of Sciences (grant no. XDA19070203) and the National Key Research and Development Program of China (grant no. 2017YFA0603802). Nadine Unger acknowledges funding support from The University of Exeter.

## Author contributions

X.Y. and N.U. designed the research and wrote the paper; X.Y. set up models and performed all simulations.

## Additional information

**Competing interests:** The authors declare no competing interests.

