## [Peer Review File · Nature Communications]

Reviewers' comments:

Reviewer #1 (Remarks to the Author):

This study assesses the indirect impact of fires on terrestrial productivity through aerosol induced changes in radiation and ozone damage on vegetation. This is the first global study combining both of these effects. Previous studies have assessed these two impacts separately for the Amazon. This study is therefore novel in expanding to the global scale and in making a combined assessment of ozone and aerosol impacts. The paper finds that damage from ozone outweighs enhancement from diffuse radiation meaning that fires have a combined net negative impact on terrestrial productivity. Overall, the paper carefully evaluates the different components of the modelling framework. It makes an important contribution to our understanding of the impacts of fire in the Earth system and will be of wide interest to a number of research communities.

I have a few major comments which I think should be addressed.

Comments

In my opinion, a weakness of the paper is a limited assessment, or acknowledgment, of the uncertainties in both the aerosol and ozone impacts. I know a full quantitative assessment of uncertainties is difficult, especially given the combination of modelling tools used. However, I think there needs to be a more complete discussion of the uncertainties, and acknowledgment that these uncertainties are considerable. The error bars in Figure 4 show interannual variability rather than an assessment of error.

An important finding is that the presence of clouds substantially reduces the impacts of fire aerosol on diffuse radiation and terrestrial productivity. The study calculates the presence of clouds reduces the GPP enhancement from fire aerosols by more than a factor 10, from 0.7 Pg C yr⁻¹ to 0.05 PgC yr⁻¹. The small response to aerosol is therefore largely due to this suppression from the presence of clouds, resulting in a net negative impact due to ozone. However, the cloud suppression is much larger than seen in previous studies. If this cloud suppression effect was smaller, as found in previous studies, the aerosol and ozone impacts would be of similar magnitude but opposite sign resulting in a net zero impact on productivity. For this reason, I think this cloud suppression requires further analysis and discussion. Are the authors convinced that their assessment is more accurate than previous studies? Is there observational evidence to support the magnitude of the cloud suppression effect simulated here? How sensitive is this suppression to uncertainty in the cloud amount, cloud optical thickness/liquid water path or other parameters?

The study uses daily cloud amount at quite coarse resolution, meaning that variability in cloud amount will be reduced, leading to an underestimation of cloud free periods. Does this impact the analysis? To help understand this, an alternative estimate of the cloud suppression would be to scale the clear sky change in GPP by the cloud fraction (i.e., for a grid cell with 70% cloud fraction, calculate the all-sky effect as 0.3 x clear sky). This basically assumes there is zero impact (neither positive or negative) of aerosol during cloudy periods. What is the GPP enhancement calculated by this method?

The diffuse impact of aerosol depends on the optical properties which in turn depends on the chemical composition of the aerosol. An evaluation of these components (including the BC/OC fraction) of the system would improve the paper.

The study excludes aerosol induced changes in meteorology, but this is probably wise given the large uncertainties associated with these changes.

Line 208-209. I think this study is at mid-latitudes so not relevant for this discussion of the Amazon.

Reviewer #2 (Remarks to the Author):

While there are several points in this manuscript that needs clarification and I have marked those on an annotated version (please see attachment), I can't stop asking myself what is novel in this manuscript to make it a Nature publication. It is relatively well known that ozone affects photosynthesis by damaging photosynthesis apparatus of leafs as it gets in through stomata. This effect (as authors say) is small and around 4 Pg C/yr and has been reported in literature before. Current global GPP is around 123 ± 8 Pg C/yr so the effect of ozone is within the uncertainty estimates of global GPP from Beer et al. (10.1126/science.1184984). The effect of ozone generated by fire is even smaller and calculated to be 0.91 Pg C/yr by authors. The same argument goes for the effect of aerosols on global GPP. I am not too familiar with literature in this respect. While the manuscript may be quantifying the effect on global GPP of ozone generated by fire alone for the first time given that the effect is so small, and then there are uncertainties on top of this, I am unsure how to use this knowledge in advancing science. Yes, the regional effects of ozone and aerosols will be larger than their globally averaged effects but then so are the uncertainties at regional scales which are affected by uncertainty in transport and chemistry processes in the GEOS-Chem model. The manuscript itself makes for a reasonable sensitivity study and as such then well worth publication in a regular scientific journal. I have always thought of Nature publications to inform the general public and scientific community of new, interesting and ground breaking knowledge. In my humble opinion, I do not feel that the science or information reported in this manuscript meets these criteria.

Responses to reviewers' comments:

Reviewer #1 (Remarks to the Author):

We are grateful to the reviewer for their time and energy in providing helpful comments and guidance that have improved the manuscript. In this response, we describe how we have addressed the reviewer's comments. Referee comments are shown in black italics and author responses are shown in blue regular text. All references cited here can be found in the revised paper.

This study assesses the indirect impact of fires on terrestrial productivity through aerosol induced changes in radiation and ozone damage on vegetation. This is the first global study combining both of these effects. Previous studies have assessed these two impacts separately for the Amazon. This study is therefore novel in expanding to the global scale and in making a combined assessment of ozone and aerosol impacts. The paper finds that damage from ozone outweighs enhancement from diffuse radiation meaning that fires have a combined net negative impact on terrestrial productivity. Overall, the paper carefully evaluates the different components of the modelling framework. It makes an important contribution to our understanding of the impacts of fire in the Earth system and will be of wide interest to a number of research communities.

→ Thank you for the positive assessment.

I have a few major comments which I think should be addressed.

Comments

In my opinion, a weakness of the paper is a limited assessment, or acknowledgment, of the uncertainties in both the aerosol and ozone impacts. I know a full quantitative assessment of uncertainties is difficult, especially given the combination of modelling tools used. However, I think there needs to be a more complete discussion of the uncertainties, and acknowledgment that these uncertainties are considerable. The error bars in Figure 4 show interannual variability rather than an assessment of error.

In the revised paper, we perform many more sensitivity experiments to quantify uncertainties in aerosol and ozone impacts (Tables S6-S9). For ozone impacts, we separate interannual variability (Fig. 5c) and uncertainties from sensitivity coefficients (Fig. 5b). We also adjust sensitivity coefficients for evergreen broadleaf forest based on observations. For aerosol effects, uncertainties in the atmospheric aerosol profile, optical parameters, and cloud amount may jointly influence our conclusions. To estimate the uncertainties of aerosol diffuse fertilization effects (DFE), in total 12 sets of runs are performed (Tables S7-S8) by assuming doubled cloud amount (C20), half cloud amount (C05), 3-hour cloud variables (C3H), no cloud (C00), doubled fire BC (BC20), half fire BC (BC05), and doubled (FA2) and tripled (FA3) total fire aerosols

(the details of these experiments will be explained in the following responses). We summarize the uncertainties in the revised Fig. 5b and discuss them in 2-4 paragraphs of Discussion section. These additional analyses help us to understand the real extent of uncertainties in the fire pollution impacts on ecosystems.

An important finding is that the presence of clouds substantially reduces the impacts of fire aerosol on diffuse radiation and terrestrial productivity. The study calculates the presence of clouds reduces the GPP enhancement from fire aerosols by more than a factor 10, from 0.7 Pg C yr⁻¹ to 0.05 PgC yr⁻¹. The small response to aerosol is therefore largely due to this suppression from the presence of clouds, resulting in a net negative impact due to ozone. However, the cloud suppression is much larger than seen in previous studies. If this cloud suppression effect was smaller, as found in previous studies, the aerosol and ozone impacts would be of similar magnitude but opposite sign resulting in a net zero impact on productivity. For this reason, I think this cloud suppression requires further analysis and discussion. Are the authors convinced that their assessment is more accurate than previous studies? Is there observational evidence to support the magnitude of the cloud suppression effect simulated here? How sensitive is this suppression to uncertainty in the cloud amount, cloud optical thickness/liquid water path or other parameters?

➔ Yes, observations suggest that dense clouds/aerosols suppress plant photosynthesis. In Yue and Unger (2017), we summarized studies of DFE (Table C1 below) for both aerosols and clouds. In the Table, we can see that plant photosynthesis reaches maximum if diffuse fraction (DF) is between 0.4-0.7 for trees. GPP reduces with heavy cloud cover (Rocha et al., 2004), and NEP decreases with AOD during cloudy days (Cohan et al., 2002). In addition, Gu et al. (2003) simulated GPP responses to cloud cover (Fig. C1) and found that GPP reaches maximum if DF is ~0.5.

Figure C1. Changes of GPP, total, diffuse, and direct solar radiation with cloud cover at Harvard Forest (adopted from Gu et al., Science, 2003).

Period	PFTs ^a	Lat.	Method	Diffusion metrics	Results ^b	Reference
1989–1990	DBF	42° S	Flux obs.	Cloud	NEP is greater on cloudy days than clear days.	Hollinger et al. (1994)
1997	Trees	>53° N	Flux obs.	Cloud	NEP is greater on cloudy days than clear days.	Law et al. (2002)
1998–2000	ENF	40° N	Flux obs.	Cloud	Maximum NEP is 11 % higher on cloudy days than clear days.	Monson et al. (2002)
1999–2001	DBF	46° N	Flux obs.	Cloud	GPP is greater under partly cloudy than clear skies, but is reduced under heavy cloud cover.	Rocha et al. (2004)
2002	ENF	39° N	Flux obs.	Cloud	Mean NEP is 7 % greater on cloudy than clear days.	Misson et al. (2005)
2001–2013	Varied	35–46° N	Flux obs.	Cloud	LUE increases with cloud optical depth (COD). GPP increases if COD < 6.8, but decreases if not.	Cheng et al. (2016)
1992–1993	DBF	43° N	Model	Cloud	Noontime GPP shows maximum increases of 40 % by cloud.	Gu et al. (2003)
1998–2002	Varied	36–71° N	Flux obs.	AOD	NEP increases with aerosol loading for forest and crop, but decreases for grassland.	Niyogi et al. (2004)
2002	ENF	39° N	Flux obs.	AOD	Afternoon NEP increases by 8 % by aerosol.	Misson et al. (2005)
1999–2002	EBF	10° S	Flux obs.	AOD	NEP increases by 29 % if AOD = 0.1–1.5.	Cirino et al. (2014)
15 July	ENF	30° N	Model	AOD	(1) NPP increases by 30 % at AOD = 0.6 for clear days. (2) NPP decreases with AOD during cloudy days.	Cohan et al. (2002)
1992–1993	DBF	43° N	Flux obs.	Radiation	Noontime GPP increases by 23 % by volcanic aerosols under clear sky.	Gu et al. (2003)
1999–2003	Trees	3–61° N	Model	Radiation	GPP falls with decreased insolation.	Alton et al. (2007)
1999–2001	DBF	46° N	Flux obs.	DF	Midday GPP is maximum at DF = 0.57.	Rocha et al. (2004)
1992–1999	Varied	1° S–71° N	Flux obs.	DF	GPP is maximum at DF = 0.4–0.7 for trees and shrubs, and DF = 0.2–0.3 for grass.	Alton (2008)
2000–2002	DBF	51° N	Flux obs.	DF	NEP is maximum at DF = 0.28–0.44.	Moffat et al. (2010)
2001–2006	Savanna	12° S	Flux obs.	DF	GPP decreases with increase in DF.	Kanniah et al. (2013)
1999–2002	EBF	10° S	Flux obs.	DF	NEP is maximum at DF = 0.6.	Cirino et al. (2014)
2001–2012	Varied	39–46° N	Flux obs.	DF	Diffuse PAR explained up to 41 % of variation in GPP in croplands and up to 17 % in forests.	Cheng et al. (2015)
2003–2013	Varied	36–46° N	Flux obs.	DF	GPP is maximum at DF = 0.4–0.6.	Strada et al. (2015)
2002	DBF	51° N	Model	DF	GPP is maximum at DF = 0.45.	Knohl and Baldocchi (2008)
2002	DBF	51° N	Model	DF	Maximum GPP enhancement of 18 % at DF = 0.4	Mercado et al. (2009)
2007	Herbs	36° N	Flux obs.	CI ^c	NEP is maximum at CI = 0.37 (DF = 0.78).	Jing et al. (2010)
2003–2006	Trees	23–36° N	Flux obs.	CI	NEP is maximum at CI = 0.4–0.6 (DF = 0.36–0.73).	Zhang et al. (2010)
2008–2009	Herbs	38° N	Flux obs.	CI	NEP is maximum at CI = 0.4–0.7 (DF = 0.18–0.73).	Bai et al. (2012)

^a Plant functional types (PFTs) include evergreen needleleaf forest (ENF), deciduous broadleaf forest (DBF), evergreen broadleaf forest (EBF), trees (mixture of ENF/DBF/EBF and shrub), herbs (grass and crop), and savanna.

^b Carbon metrics include gross primary productivity (GPP), net primary productivity (NPP), and net ecosystem productivity (NEP).

^c Clearness index (CI) is converted to diffuse fraction (DF) with $DF = 1.45 - 1.81 \times CI$ (Alton, 2008).

Table C1. Summary of studies about diffuse fertilization effect (DFE) (adopted from Yue and Unger, ACP, 2017).

➔ Theoretically, for each location on the Earth, a specific cloud amount leads to optimized GPP. It is important to estimate this threshold so as to quantify the potential range for aerosol DFE. In the revised paper, we performed 20 sensitivity experiments with cloud scaling factors from 0 to 10 without aerosol radiative effects (Tables S4-S5). This method is different from Gu et al. (2003) which uses simplified relationships between radiation and cloud cover. In our radiation model, we use 3-D cloud cover and liquid water path from observations. The scaling method will apply a factor between 0 and 10 to both cloud cover and liquid water path so as to retain the observed cloud profile but with the altered amount/magnitude. GPP from each simulation is compared with others to identify the maximum GPP and derive the corresponding cloud scaling factor. We plot global GPP responses to diffuse and direct radiation in Fig. S3 which shows that GPP reaches the maximum at cloud scaling of 0.9, suggesting that ecosystem productivity is in general already almost optimized at current cloud amount. Fig. S3 is different from Fig. C1 because we looked at global GPP with observed cloud profile using a radiative transfer model, while Gu et al. (2003) used simplified cloud-radiation relationship at the Harvard Forest site.

➔ With this cloud scaling method, we derived a global map for cloud DFE threshold (Fig. 3). We can see that thresholds of cloud scaling are usually < 1 for boreal and

tropical forest, suggesting that cloud is very dense there and the optimal GPP can be achieved only if current cloud amount is reduced. As a result, the dense cloud in boreal and tropical forest will mask aerosol DFE, leading to limited aerosol effects on plant photosynthesis. On the other hand, thresholds of cloud scaling are higher than 1 over arid and semi-arid regions, suggesting that GPP is not optimized with the current cloud amount and aerosols can further increase photosynthesis in those regions. It is important to recognize that the cloud masking threshold is not strictly equal to 1. For some regions where thresholds of cloud scaling are between 0.8-1, aerosol DFE can be still positive, likely because aerosols in dry seasons (less cloud) increase GPP (the mismatch between cloud and aerosol seasons). Fig. 3 shows us where on the Earth aerosol DFE has the potential to increase annual GPP.

→ Direct observations of cloud impacts on photosynthesis are limited (see Table C1) but very difficult to quantify, not only because cloud amount and properties are hard to determine, but also that changes of photosynthesis are jointly affected by environmental factors in addition to diffuse radiation. For this study, we carefully validated modeled DFE (no matter by clouds or aerosols) for YIBs model using observations from 24 FLUXNET sites (Fig. 2 and Figs 1-2). We applied the bin method proposed by Mercado et al. (Nature, 2009) so as to minimize impacts of other environmental factors. With these evaluations, we have confidence in the simulated GPP responses to cloud and aerosol DFE. We also performed several sensitivity experiments to quantify how the uncertainties in cloud variables will affect the aerosol DFE from fires (see following responses).

The study uses daily cloud amount at quite coarse resolution, meaning that variability in cloud amount will be reduced, leading to an underestimation of cloud free periods. Does this impact the analysis? To help understand this, an alternative estimate of the cloud suppression would be to scale the clear sky change in GPP by the cloud fraction (i.e., for a grid cell with 70% cloud fraction, calculate the all-sky effect as 0.3 x clear sky). This basically assumes there is zero impact (neither positive or negative) of aerosol during cloudy periods. What is the GPP enhancement calculated by this method?

→ The reviewer is interested in how the spatial and temporal resolutions of the cloud data products affect the results. We perform additional experiments to check the impact of the temporal resolution on cloud and aerosol DFE (Tables S7-S8). In the first test “C3H”, the daily cloud variables (cover and liquid water path) are updated with 3-hour data from CERES. Simulated Δ GPP by fire aerosols with 3-hour cloud is 0.01 Pg C yr⁻¹, slightly lower than the value of 0.05 Pg C yr⁻¹ with daily cloud.

→ All simulations are performed at 1° spatial resolution because cloud variables from SYN1deg product are available at maximum 1° resolution. This coarse spatial resolution will affect the derived aerosol DFE. The reviewer suggests that we can assume a grid cell with 70% cloudy sky and 30% clear sky. The derived GPP

enhancement by fire aerosols is $0.25 \text{ Pg C yr}^{-1}$ with this method ($=0.3 \times \Delta\text{GPP}_{\text{clear}} + 0.7 \times \Delta\text{GPP}_{\text{cloud}}$), but the total cloud amount is reduced by 30% as a result. To reserve the global total cloud amount, we perform additional simulations with a slightly different strategy. We divided each grid box into 4 cells and assumed randomly 1 out of 4 cells is clear sky at each time step (25% clear-sky is similar to the ratio of 30%). To keep the conservation of total cloud amount, the other 3 cells will have 1.33 times of cloud variables at the same moment (so $(0 \times 1 + 1.33 \times 3) / 4 = 1$). We perform 4 runs to check whether the randomness affect the final result. Simulated ΔGPP by fire aerosols is $0.12 \text{ Pg C yr}^{-1}$ for all these runs. This value is higher than the $0.05 \text{ Pg C yr}^{-1}$ with observed 1° cloud profiles but lower than the value of $0.25 \text{ Pg C yr}^{-1}$ assuming 70% cloudy sky. Results of these runs are not shown in the revised text as we perform following simulations to cap the uncertainty range due to clouds.

- We consider the largest resolution limitation to be caused by the monthly fire emissions data product that may cause some potential inconsistency between fire events and cloud status. The ideal situation would be to have simultaneous hourly cloud and fire aerosols at the global-scale, but such high resolution data products are not available. We performed two sensitivity runs to cap the range caused by fire-meteorology interactions. The “C05” run assumes that cloud fraction and liquid water path is less by 50%, because fires tend to occur in dry moment when cloud amount is relatively low, and fire aerosols can inhibit cloud and precipitation through fire-meteorology interactions (Liu et al., 2014). In contrast, “C20” run assumes that cloud fraction and liquid water path increase by 100%, because other studies (e.g., Lu et al., 2017) found that fire aerosols will increase cloud amount and depth through aerosol indirect effects. Fire aerosols in “C05” cause a much stronger fertilization effect of $0.40 \text{ Pg C yr}^{-1}$ to global GPP, while in “C20” cause an inhibition of $-0.13 \text{ Pg C yr}^{-1}$. As a result, we use -0.13 and 0.40 as the uncertainty range (Fig. 5b) for fire aerosol DFE relative to cloud amount. Related discussion can be found in the revised text (Lines 265-275)

The diffuse impact of aerosol depends on the optical properties which in turn depends on the chemical composition of the aerosol. An evaluation of these components (including the BC/OC fraction) of the system would improve the paper.

- Yes, aerosol composition and optical properties will affect fire aerosol DFE. First, we performed two simulations with doubled (FA2) and tripled (FA3) fire aerosols (Tables S7-S8). Predicted ΔGPP in these two simulations are respectively $0.06 \text{ Pg C yr}^{-1}$ and $0.04 \text{ Pg C yr}^{-1}$, very similar to the value of $0.05 \text{ Pg C yr}^{-1}$ with the original fire emissions. Second, fire aerosols are dominated by organic (OC) and black carbon (BC), which have distinct optical properties. Two sensitivity runs are performed with different loading of BC. Simulation “BC05” assumes fraction of BC in fire pollution is reduced by 50% while OC increases by 5%, because emission factor of BC is nearly 1/10 of that for OC in fires. As a comparison, simulation

“BC20” assumes the loading of BC is doubled while OC decreases by 10%. Predicted diffuse fraction is similar between BC05 and BC20 (Table S7), but the total PAR is smaller in BC20 because BC is more absorptive than OC. Consequently, Δ GPP by fire aerosols in BC05 is 0.10 Pg C yr⁻¹, higher than the value of -0.06 Pg C yr⁻¹ in BC20. In sum, uncertainties in fire aerosols drive varied GPP responses from -0.06 to 0.10 Pg C yr⁻¹, much smaller than those of -0.13 to 0.40 Pg C yr⁻¹ due to changed cloud properties (Fig. 5b). Related discussion can be found in the revised text (Lines 277-292)

The study excludes aerosol induced changes in meteorology, but this is probably wise given the large uncertainties associated with these changes.

➔ We agree with the reviewer. For this study, we quantified only aerosol DFE on GPP. The FLUXNET data provide a good source of evaluation of DFE so as to reduce modeling uncertainties. In the future, we consider to include aerosol feedbacks to climate (changes in temperature and precipitation) and the consequent impacts on ecosystem productivity.

Line 208-209. I think this study is at mid-latitudes so not relevant for this discussion of the Amazon.

➔ We understand and agree with the reviewer’s point, however, the only two studies that exist for any comparison are focused on the Amazon. We would like to compare/contrast our results with theirs and discuss the causes of discrepancies.

Reviewer #2 (Remarks to the Author):

We are grateful to the reviewer for their time and energy in providing helpful comments and guidance that have improved the manuscript. In this response, we describe how we have addressed the reviewer's comments. Referee comments are shown in black italics and author responses are shown in blue regular text.

General comments

While there are several points in this manuscript that needs clarification and I have marked those on an annotated version (please see attachment), I can't stop asking myself what is novel in this manuscript to make it a Nature publication. It is relatively well known that ozone affects photosynthesis by damaging photosynthesis apparatus of leafs as it gets in through stomata. This effect (as authors say) is small and around 4 Pg C/yr and has been reported in literature before. Current global GPP is around 123 ± 8 Pg C/yr so the effect of ozone is within the uncertainty estimates of global GPP from Beer et al. (10.1126/science.1184984). The effect of ozone generated by fire is even smaller and calculated to be 0.91 Pg C/yr by authors. The same argument goes for the effect of aerosols on global GPP. I am not too familiar with literature in this respect. While the manuscript may be quantifying the effect on global GPP of ozone generated by fire alone for the first time given that the effect is so small, and then there are uncertainties on top of this, I am unsure how to use this knowledge in advancing science. Yes, the regional effects of ozone and aerosols will be larger than their globally averaged effects but then so are the uncertainties at regional scales which are affected by uncertainty in transport and chemistry processes in the GEOS-Chem model. The manuscript itself makes for a reasonable sensitivity study and as such then well worth publication in a regular scientific journal. I have always thought of Nature publications to inform the general public and scientific community of new, interesting and ground breaking knowledge. In my humble opinion, I do not feel that the science or information reported in this manuscript meets these criteria.

→ **The importance of this study is not measured by the magnitude of GPP response.** First, we reveal a missing mechanism for carbon loss caused by fire air pollution. On the global scale, the loss is -0.85 Pg C every year, suggesting that global forests may absorb 85 Pg less carbon in a century. Such accumulation effect will definitely influence atmospheric CO₂ concentrations and the consequent global warming tendency. Second, regional perturbations can be as high as -3.6% at current climate. In a warmer future, wildfire is predicted to increase with higher intensity, suggesting that fire air pollution may have worse effects then. Third, fire pollution-induced GPP changes are not small compared with other factors. For example, drought reduces global NPP by 0.55 Pg C in a decade (Zhao and Running, Science, 2010), while fire pollution reduces global GPP by 0.85 Pg C per year. In addition, a recent study shows that biogenic aerosol enhances global GPP by 1.26 Pg C yr⁻¹ through SOA diffuse fertilization (Rap et al., Nature Geoscience, 2018), which is of the comparable magnitude to fire pollution effects but with opposite sign.

- **The uncertainties of fire pollution effects have been fully evaluated.** In the revised text, we perform many more sensitivity experiments to explore modeling uncertainties (Tables S6-S9). We summarize the uncertainties in the revised Fig. 5. As it shows, for some effects, modeling uncertainties are larger than the net effects (e.g., fire aerosol DFE). However, this does not mean such effect is trivial. For example, radiative forcings of several aerosol species (e.g., biomass burning aerosols and SOA) are near zero with large uncertainties (Fig. C2 below). But the IPCC Working Groups I, II and III communities all believe that these species are important for the radiation budget. We agree that the predicted $0.85 \text{ Pg C yr}^{-1}$ loss by fire pollution is indeed much smaller than the 123 Pg C yr^{-1} of global GPP. Similarly, global aerosol radiative forcing is -1.5 W m^{-2} , much smaller than the average solar radiation of 190 W m^{-2} at the surface (Fig. S14 in the SI). However, we cannot say aerosol radiative perturbations are not important.
- **Findings of this study are novel and scientifically robust.** We agree that Nature journals have very high standards, which are the targets for us to improve this work. The idea of fire pollution impacts on ecosystem productivity is the frontier of pollution-biosphere interaction researches. For example, Rap et al. just published “Enhanced global primary production by biogenic aerosol via diffuse radiation fertilization” in Nature Geoscience on August 20th, 2018. We can see that the topic of aerosol DFE is attracting large attention in the broad scientific community. Compared to Rap et al., this work further advances pollution-biosphere interactions by analyzing both ozone damaging and aerosol DFE from fire pollution (and related uncertainties), and most important, provides extensive evaluations of ozone effects (Fig. 1), aerosol DFE (Fig. 2), global GPP (Figs. S11-S12), pollution concentrations (Fig. S13), radiation flux (Fig. S14), and aerosol radiative effects (Fig. S15).

Figure C2. Annual mean top of the atmosphere radiative forcing due to aerosol–radiation interactions by different anthropogenic aerosol types for 1750–2010 (adopted from IPCC AR5 report Figure 7.18).

Specific comments:

(1) Line 20, “Without fire emissions, O₃ decreases ...” What is the source of this O₃?

→ This O₃ includes all natural and anthropogenic sources except fires, In the text, it has been revised as "With all emissions except fires, O₃ decreases ..." (Line 20)

(2) Lines 24-25, “The net negative impact of fire pollution poses an increasing threat to ecosystem productivity in a warming future world.” Seems overstatement.

→ The net effects of fire pollution on ecosystem will accumulate and amplify in a warmer climate, leading to substantial impacts to global productivity. (Please refer to our main responses to general comments).

(3) Lines 34-36, “Regionally, fires ... dominate PM_{2.5} with average contributions of 30-55% during fire seasons.” Does this mean during fire 30-55% aerosols in the near surface air are from fire?

→ Yes. For example, the Ref 8 (Fujii et al., 2017) shows that Indonesia fires account for 51-55% of total PM_{2.5} locally during dry season.

(4) Page 3, what is current O₃ concentration?

→ Current O₃ concentration is shown and compared with observations in Figure S13b.

(5) Fig. S2: Does ‘data points’ mean observations?

→ Yes, 'data points' mean observations. It has been revised as "The categorized observations ..." to clarify.

(6) Line 64, “literature-based meta-analysis ...” are these model or observations?

→ These are observations. In the text, it has been revised as "observation-based meta-analysis"

(7) Line 68, the word “Nevertheless” should be deleted.

→ Corrected as suggested.

(8) Line 76, “GPP grow faster” add “with light”.

→ Corrected as suggested.

(9) Page 4, Fig S3: Also show background O₃ concentration and AOD without fire.

→ Background O₃ and AOD without fires have been shown in Figures S5c and S5d in the revised paper.

(10) Lines 87-89, “The largest enhancement of [O₃] by >15 ppbv is found in southern Africa (Fig. S3a), where fires on average contribute 6.9 ppbv (21.3%) to the local [O₃].” Can’t reconcile the numbers of 15 and 6.9 ppbv.

→ 15 ppbv is the maximum at 1-2 grids while 6.9 ppbv is the average value over southern Africa. In the text, it has been revised as “The largest enhancement of [O₃] (grid maximum >15 ppbv) is found in southern Africa”.

(11) Line 91, “on the annual mean basis” here “the” should be “an”

→ Corrected as suggested.

(12) Line 94, “However, for these grids ...” you mean Siberia/Alaska or global?

→ These grids are 2/3 land grids mentioned in the previous sentence. It is revised as follows: “Globally, fires enhance surface O₃ by at least 0.5 ppbv over 2/3 land grids, for which we find a negative correlation of -0.36 ...”

(13) Lines 97-98, “In addition to O₃, fires contribute to regional AOD by 39.2% in southern Africa, 33.0% in tropical Amazon, and 31.8% in Indonesia” Does this mean annually this % is attributed by fires?

→ Yes. It is clarified by adding “on an annual mean basis” at the end.

(14) Lines 105-106: “Without any fire emissions, global GPP is reduced by 4.0 ± 1.9 Pg C yr⁻¹ (2.9 ± 1.4 %, mean \pm uncertainties) due to O₃ effect” compared to what?

→ We clarified as follows: “With all emissions except from fires, global GPP is reduced by 4.0 ± 1.9 Pg C yr⁻¹ (2.9 ± 1.4 %, mean \pm uncertainties) due to O₃ damage losses compared to O₃-free GPP” (Lines 127-128)

(15) Lines 111-112: “enhancement of O₃ causes additional GPP reductions of 1.7% in central Africa, 1.5% in Indonesia, 0.5% in central South America, and 0.6% in northeastern Asia during 2002-2011” These are overall small changes.

→ First, these values are multi-year averages. For large fire years, e.g., fire O₃ reduces GPP by 3.6% over Indonesia in 2006. Second, fire-free O₃ reduces global GPP by 2.9%. The key point of these results is that fire perturbations cause GPP reductions of comparable orders to fire-free pollution. Finally, such damage effects happen every year, and likely enhance in a warmer climate with more fires.

(16) Page 5, Fig. S4 is difficult to understand. What does “Change > 80th percentile” mean?

→ Fig. S4 now becomes Fig. S6 in the revised text. This figure and the corresponding text mainly explain why over some regions large fire O₃ causes small changes in GPP, and why in other areas low fire O₃ causes large perturbations in GPP. The main cause for such apparent discrepancy is related to ambient O₃ concentration (determining the baseline) and stomatal conductance (determining uptake rate). In the revised text, we offer a more clear explanation of Fig. S6 including an updated caption. Specifically, we clarify how O₃ damage is dependent on both the stomatal conductance and O₃ concentration in the Methods section.

(17) Lines 117-118: “suggesting that the higher stomatal conductance drives the larger O₃ uptake”

→ Based on Sitch's scheme, O₃ damage is dependent on stomatal flux, which relies on both [O₃] and stomatal conductance (Methods, equations 3 and 4). For the same [O₃], the higher stomatal conductance, the larger stomatal flux. Here, we found that over some regions with high stomatal conductance, GPP damage is usually high even with moderate [O₃].

(18) Lines 120-125, this point needs to be brought out more clearly.

→ For this paragraph, we summarize our conclusion in the first sentence as follows: "Large fire Δ [O₃] does not always cause high GPP reductions, and vice versa."

(19) Line 134, “cloud fraction” Does cloud cover comes from GEOS-Chem model? Has it been evaluated?

→ The cloud cover is from satellite-based observations. In the Methods section, we explained as follows: “The CRM is driven with ... daily cloud profile (e.g., cloud fraction and liquid water path) from CERES SYN1deg assimilation data product (<http://ceres.larc.nasa.gov>), which is developed based on satellite retrievals from the Moderate Resolution Imaging Spectroradiometer (MODIS) and the Visible and Infrared Sounder (VIRS).” (Lines 332-337)

(20) Line 137, “with the inclusion of cloud effects” how?

→ The CRM model uses observed cloud profile as input. In one sensitivity experiment, all cloud variables are set to 0 (C00, Table S7). In the other experiment, cloud variables from observations are used (C10, Table S7). Then the derived GPP with corresponding PAR are compared.

(21) Line 141, “former ... regions” what does former mean here?

→ We clarified as follows: “...over the arid/semi-arid regions with $\Delta\text{GPP} > 2\%$ ” (Line 169)

(22) Lines 162-163: “Plants at high latitudes have much larger shaded portion than that at low latitudes (Methods).” Unclear. Why? Due to angle of sun?

→ Yes. The main cause is that the small solar zenith at high latitudes results in more shading portion of leaves (see equation 2 in the Methods). We clarified as follows: “Plants at high latitudes have much larger shaded portion than that at low latitudes because of the smaller solar zenith at high latitudes”

(23) Line 182, “even with regional offsets” unclear.

→ This conclusion is adopted from van der Werf et al. (2010): “On a regional basis, emissions were highly variable during 2002–2007 (e.g., boreal Asia, South America, and Indonesia), but these regional differences canceled out at a global level.

(24) Lines 185-188, “On average, the interannual variation (one standard deviation) accounts for 5% of the mean responses in GPP to fire pollution, much smaller than the uncertainties of around 43% driven by O_3 damage sensitivity (low or high).” Unclear.

→ This sentence compares interannual variation of fire pollution effects with uncertainties in ozone damage effect due to low/high ozone sensitivity parameters. To clarify, we revised this sentence as follows: “On average, the interannual variation ($0.05 \text{ Pg C yr}^{-1}$, one standard deviation for 2002-2011) accounts for 5% of the mean ΔGPP by fire pollution, much smaller than the uncertainties of 43% driven by O_3 damage sensitivity ($0.37 \text{ Pg C yr}^{-1}$, half of the range from low to high sensitivities).”

(25) Lines 188-193, these are overall very small changes.

→ Both the regional and global perturbations in GPP by fire pollution are not trivial. Please see our responses to the general comments.

(26) Line 211, “relatively high responses (0.2-0.4%)” Again these are overall very small numbers.

→ This response refers to a specific location instead of the global impact by fire pollution. In other regions, the fractional change number is much higher.

(27) Lines 239-240, say more here about ref #37.

→ We revised the sentence as follows: “As a comparison, drought is estimated to reduce net primary production by 0.55 Pg C (1%) during 2000-2009⁴¹. The fire pollution-induced inhibition (0.6% per year) is much stronger than the drought effect (0.1% per year) for the same period.” Notice that the Ref. 37 is now Ref. 41.

(28) Lines 242-244, “Such perturbations may result in the loss of land carbon storage and exacerbate the global warming trend due to increasing the atmospheric CO₂ burden.” This is likely an overstatement.

→ The quantitative results mean that this sentence is not an overstatement. It may actually be an understatement for the future world conditions, for instance, at least to the extent that the carbon cycle community is concerned about drought impacts on productivity and land carbon uptake/storage. The net effects of fire pollution on ecosystems (how burning vegetation impacts non-burning vegetation) will accumulate and amplify in a warmer climate, leading to substantial impacts on global productivity. Current assessments of the fire contribution to the carbon-climate feedback have been underestimated because they do not consider the fire pollution feedback (e.g. Harrison et al., The biomass burning contribution to climate-carbon-cycle feedback, *Earth Syst. Dynam.*, 9, 663-677, <https://doi.org/10.5194/esd-9-663-2018>, 2018). Please see the responses to general comments.

(29) Line 254, I think the emissions of NMVOCs are specified and then the model simulates the interactions.

→ For anthropogenic NMVOCs, yes, the emissions are specified. However, for biogenic VOCs such as isoprene and monoterpenes, the GC model simulates emissions using MEGAN scheme based on temperature and radiation.

(30) Line 262, “predicts” should be “simulates”.

→ Corrected as suggested.

(31) Lines 298-300, “Globally, satellite-based retrievals of 8 PFs are applied in the YIBs model”. What does this mean?

→ We clarified as follows: “The model considers 8 PFTs, including evergreen needleleaf forest, deciduous broadleaf forest, EBF, shrubland, tundra, C3/C4 grass, and C3 crops. The satellite-based land types and cover fraction⁴⁸ are aggregated into these 8 PFTs and used as input.”

(32) Line 319, “All simulations are performed for 2001-2011” add “repeatedly”.

→ Thank you for the suggestion, but the word “repeatedly” might be misleading as all

simulations are performed for the same decadal temporal period but with different configurations and settings. As a result, we did not use it here.

(33) Line 321, “*area-weighted downscaled*” change “*area-weighted*” to *conservatively*?

→ Changed as suggested.

(34) Line 323, “*interpolated*” change to “*regridded*”

→ Changed as suggested.

(35) Figure 2 caption: “*Both observations (red) and simulations (blue) are calculated ...*” English! What does “*observations are calculated*” mean?

→ We changed the statement as follows: “The changes of GPP are calculated for 24 FLUXNET sites (Fig. S1) using both observed (red) and simulated (blue) data.”

REVIEWERS' COMMENTS:

Reviewer #1 (Remarks to the Author):

The authors have added an impressive range of sensitivity studies and model evaluation to address reviewer comments. The new analysis provides a more comprehensive analysis of uncertainties and the sensitivity to ozone and aerosol.

The authors have addressed all my comments. Overall, I think the revised manuscript makes an important and novel contribution to our understanding of the impacts of fire in the Earth system. In my opinion the revised manuscript is suitable for publication In Nature Communications.